# Role of Cytokines in Vitiligo: Pathogenesis and Possible Targets for Old and New Treatments

**DOI:** 10.3390/ijms222111429

**Published:** 2021-10-22

**Authors:** Paolo Custurone, Luca Di Bartolomeo, Natasha Irrera, Francesco Borgia, Domenica Altavilla, Alessandra Bitto, Giovanni Pallio, Francesco Squadrito, Mario Vaccaro

**Affiliations:** 1Department of Clinical and Experimental Medicine, Dermatology, University of Messina, Via C. Valeria, Gazzi, 98125 Messina, Italy; paolo.custurone@gmail.com (P.C.); lucadibartolomeo@live.it (L.D.B.); francesco.borgia@unime.it (F.B.); 2Department of Clinical and Experimental Medicine, Pharmacology, University of Messina, Via C. Valeria, Gazzi, 98125 Messina, Italy; natasha.irrera@unime.it (N.I.); domenica.altavilla@unime.it (D.A.); alessandra.bitto@unime.it (A.B.); giovanni.pallio@unime.it (G.P.); francesco.squadrito@unime.it (F.S.)

**Keywords:** vitiligo, interleukins, biologics, biological drugs, oxidative stress, autoimmune diseases, depigmentation, dermatoses, inflammation, skin

## Abstract

Vitiligo is a chronic autoimmune dermatosis of which the pathogenesis remains scarcely known. A wide variety of clinical studies have been proposed to investigate the immune mediators which have shown the most recurrency. However, such trials have produced controversial results. The aim of this review is to summarize the main factors involved in the pathogenesis of vitiligo, the latest findings regarding the cytokines involved and to evaluate the treatments based on the use of biological drugs in order to stop disease progression and achieve repigmentation. According to the results, the most recurrent studies dealt with inhibitors of IFN-gamma and TNF-alpha. It is possible that, given the great deal of cytokines involved in the lesion formation process of vitiligo, other biologics could be developed in the future to be used as adjuvants and/or to entirely replace the treatments that have proven to be unsatisfactory so far.

## 1. Introduction

Vitiligo is a common chronic autoimmune dermatosis, mainly characterized by milky white patches with distinct edges, affecting 1% of the population worldwide [1]. Two clinical forms may be recognized, segmental and non-segmental vitiligo, with the unilateral distribution of the patches and an earlier onset in the segmental variant and a scattered distribution in the non-segmental form [1]. Both clinical manifestations are considered as a continuous spectrum with the same pathologic process, although segmental vitiligo responds to the theory of cutaneous mosaicism [2], which means that two separate cell lines coexist side by side in the skin in the same patient. The most common variant is non-segmental vitiligo, which is characterized by white macules/patches scattered over the body, in particular skin folds and areas around the mucosae (mouth, orbits, genitals, anus, wrists, fingertips), progressing in time in the extent and number of patches, with a symmetrical distribution. White patches usually appear in visible areas which are difficult to cover; for this reason, vitiligo affects the quality of life of patients [3]. The appearance of vitiligo is often associated with autoimmune diseases, such as Hashimoto’s thyroiditis [4]; in fact, both high levels of CD8+ cytotoxic lymphocytes in lesions and autoantibodies have been found [5]. Over time, a great variety of factors, such as inflammatory products, reactive oxygen species, trauma, sunlight exposure and genetic factors, may play a significant role in vitiligo, and therefore might be considered potential targets for future and present therapies. The aim of this literary review was to provide a summary of the pathogenetic processes of vitiligo, thus focusing on the factors involved and the possible therapeutic approaches to use in clinical practice.

## 2. Vitiligo Pathogenesis: Ab Origine Development, from Gene to Cellular Dynamics

### 2.1. Predisposing Factors

Several pathogenetic processes have been proposed to describe the pathogenesis of vitiligo. Among the predisposing factors, genetics seems to play an important role [6]. In this context, the human leukocyte antigen (HLA) cluster codifies for the antigen presenting-proteins and, even though it appears not to be involved with vitiligo, several alleles appear to be co-inherited in the affected patients, some of which are linked to other autoimmune/autoinflammatory diseases, such as rheumatoid arthritis and diabetes *HLA-DRB4*0101* and *HLA-DQB1*0303* [7]. Furthermore, melanocyte proliferating gene 1 (*MYG1*), specifically the -119G variant which produces a protein (Myg1), is involved in the pathogenesis since it is indispensable for mitochondria [8] in melanocytes. Several other genes have been proposed as a possible cause of vitiligo, such as genes involved in the immune response (cytotoxic T-lymphocyte antigen 4, Fas ligand, TNF superfamily member 11), interleukins and the complement system (*IL1RAPL1, C1QTNF6*), and the gene that encodes for the tyrosinase (*TYR*), an autoantigen specific to melanocytes [9]. As of today, no gene therapies have been proposed. On a further level, epigenetic processes have also been suggested as possible pathogenetic causes. The role of micro-RNAs (miRNA), small strings of non-encoding RNA that function as silencers and gene expression regulators, have also been described in the pathogenesis of vitiligo. Several miRNAs have been recognized over time, for example, increased miR-196 and miR-3940 levels were observed in patients affected by vitiligo. Polymorphisms involving miR-196 may play a significant role in the pathogenesis and in the prognosis of the disease, suggesting its possible use as a marker or therapeutic target for future treatments [10]. Although some miRNAs, such as miR-21, are important in the normal melanogenesis process, an increase in affected patients was observed compared to healthy controls, possibly because of the loss of pigment in skin lesions [11]. On a more macroscopic level, miRNAs are linked to oxidative stress as in other diseases [12]. To overcome this stress environment, several enzymes and products are involved: vitamins C and E, glucose-6-phosphate dehydrogenase, superoxide dismutase and glutathione peroxidase, along with several catalase enzymes (responsible for hydrogen peroxide production) [13,14]. The accumulation of H_2_O_2_, as an oxidative stress product, represents the cause of the disruption of mechanisms in melanocytes, such as the alteration of calcium homeostasis, adrenocorticotropic hormone (ACTH) and melanocyte stimulating hormone (MSH) inhibition and lipid peroxidation [15,16,17]. In fact, antioxidative enzymes, namely, heme oxygenase-1 (HO-1) and the nuclear factor E2-related factor 2–antioxidant response element (Nrf2-ARE) are involved in a negative feedback pattern in the presence of H_2_O_2_ [18] and are protective against oxidative stress processes. In addition to antioxidant enzymes, autophagy is also involved in reducing reactive oxygen species (ROS). Nitric oxide (NO) is a reactive oxygen species involved in different processes, such as immunomodulation and inflammation [19]. Increased NO levels and the inducible NO synthase (iNOS) were detected in both melanocytes and keratinocytes; moreover, NO may induce melanocyte self-destruction in vitiligo [20], thus demonstrating that this product plays an important role in the pathogenesis of vitiligo [21]. On the same note, advanced glycosylation end-products (AGE) and advanced oxidation protein products (AOPP) have also been postulated as markers of disease activity and progression. These products represent the end point of oxidative processes and act as a stimulus for the immune system to turn its attention towards specific bodily cells (in this case melanocytes), leading to inflammation and killing phenomena [22]. Finally, another critical aspect in the pathogenesis of vitiligo is represented by physical trauma (responsible for Koebner’s phenomenon) and several studies have been conducted to investigate the role of integrins. An overall reduced expression of integrins was observed in both healthy and affected skin of vitiligo patients according to a 2017 study which demonstrated that beta-1 integrin, laminin, VCAM-1 and ICAM-1 are lacking in epidermis [23] close to the basal layer, where melanocytes reside. Another protein, melanoma inhibitory activity (MIA), which acts as a ligand to integrins, could be the cause of melanocytes “silently” leaving the epidermis, which translates to a possible metastatic process in melanoma and the formation of white patches in vitiligo [24]. All the above-mentioned mechanisms are not separated but rather are linked to each other. In a study by Mengyun et al. the correlations between levels of miR-9, IL-10, E-cadherin, and beta1-integrin were evaluated after UV exposure by phototherapy, suggesting a web of interactions between such molecules at a very fine and complex level [25]. All these triggering factors, obviously, are followed shortly by the activation of the immune cells. Here, the Th1 subset of the CD4+ lymphocytes and the consequential, fundamental production of various interleukins play major roles.

### 2.2. Role of Cytokines in Vitiligo

Vitiligo is characterized by an imbalance between regulatory T (Treg) and effector T cells (Teff), with impaired Treg function and increased proliferation of CD8+ and CD4+ T cells [26]. This immune activation is the last step of several phases of the pathogenetic process of vitiligo. Melanocytes are more sensitive to oxidative stress in vitiligo [27] and the increased ROS production could result from an external stress, such as ultraviolet (UV) radiation exposure or chemical damage. Stressed and damaged melanocytes and keratinocytes release damage-associated molecular patterns (DAMPs), which represent signals of damage and are recognized by pattern recognition receptors (PRRs), expressed by several types of innate immune cells, such as dendritic cells [28]. Innate immune cells express different PRRs, including Toll-like receptors on cell membrane and NLRs in the cytoplasm [29]. Some NLRs, such as NLRP1 and NLRP3, are activated in response to oxidative stress or cellular damage in vitiligo [29,30]; NLRP1 and NLRP3 are components of the inflammasome complex that, once activated, releases caspase 1 and IL-1β and IL-18 [29,30]. The inflammasome is activated in perilesional keratinocytes of vitiligo and promotes the cutaneous T cell response [29]. At the same time, other inflammatory cytokines, such as TNFα and INFγ, increase in response to cellular damage and are involved in the progression of vitiligo. The role of the main cytokines involved in the pathogenesis of vitiligo is described in the following sections.

#### 2.2.1. INFγ

Interferon-gamma is a cytokine that is mainly involved in proliferation inhibition, apoptosis and immunomodulation processes [31]. Both INFγ and TNFα are overexpressed in vitiligo [32]. INFγ is strictly involved in the development of the disease but TNF-α does not seem to have the same role, as demonstrated in TNFα-knockout mice, which are not protected by the development of vitiligo [33]. INFγ expression is associated with disease activity: in Smyth line chickens (SLC), an animal model of vitiligo, INFγ levels are increased during active phases [34]. INFγ plays several roles in melanocytes: it inhibits melanogenesis, increases ROS release and induces their senescence and apoptosis by CD8+ cells [35,36]. The cellular response to INFγ is mediated by the Janus kinase (JAK)-signal transducer and activator of transcription (STAT) pathway [37]. Both JAK1 and JAK3 are upregulated in perilesional and lesional vitiligo skin compared to the levels observed in controls [38] and the effects of INFγ on melanocytes, such as their senescence, may be attenuated through STAT1 inhibition [36]. STAT1 signaling may be involved in the inhibition of T_reg_ by IFNγ [39]. Nevertheless, STAT1 is not the only transducer involved in vitiligo, but STAT3 is also increased in vitiligo lesions [40]. INFγ plays an important role in the exacerbation of inflammation and in the interplay of keratinocytes and lymphocytes. IFNγ promotes skin homing of melanocyte-specific CD8+ cytotoxic T lymphocytes (CTLs), inducing the production of several chemokines by keratinocytes, in particular CXCL10 [41,42]. The interaction between CXCL10 and its receptor on autoreactive T cells, CXCR3, is involved in the maintenance of depigmentation in vitiligo [41] and the expression of CXCL10 correlates with the severity of the disease [42]. In conclusion, IFNγ and its downstream pathway seems to be one of the most interesting candidates for target therapy.

#### 2.2.2. TNFα

TNFα activity is increased in active vitiligo lesions [43]. TNFα induces melanocyte dysfunction and cellular death through several mechanisms [43,44]: it produces its effects acting upon the microphthalmia-associated transcription factor (MITF), melanocyte-stimulating hormone receptor (MSH-R) activity and reducing the expression of melanocortin-1 receptor (MC1-R), thus inducing melanocyte toxicity and enhancing ROS production [43,44]. Nevertheless, this pro-inflammatory cytokine appears to also play a protective role in vitiligo, activating and promoting the development of T_reg_ cells [45]. For that reason, no definitive conclusion can be drawn on a clear role of this molecule in vitiligo.

#### 2.2.3. IL-33

IL-33 is a cytokine of the IL-1 family and induces the production of Th2 cytokines [46]. IL-33 is produced in keratinocytes stimulated with both TNFα and IFNγ [46]. IL-33 serum levels are more elevated in vitiligo patients than in controls [46,47]. In patients with vitiligo, IL-33 is transferred from the nucleus to the cytoplasm in keratinocytes [46]. In vitiligo, IL-33 may be considered an alarmin; it may be released by apoptotic or necrotic keratinocytes and inhibits melanocytes growth, blocking growth factors and increasing pro-inflammatory IL-6 and TNFα expression.

#### 2.2.4. IL-1β

This cytokine is one of the main products of the inflammasome, which has a primary role in initiating inflammatory response in vitiligo, as mentioned above. Both nucleotide-binding oligomerization domain, leucine rich repeat and pyrin domain containing 1 (NLRP1) and the NLRP3 inflammasome have been reported to act in the pathogenesis of vitiligo. NLRP1 has been found to be strongly positive in progressing margins of vitiligo and NLRP1 and IL-1β are significantly associated with progressive disease rather than stable vitiligo [48,49,50]. IL-1β reduces MITF-M mRNA expression, a factor that stimulates the expression of melanocyte differentiation genes [51]. Decreased MITF levels have been found in lesional and perilesional vitiligo samples [49]. Oxidative stress promotes the expression of NLRP3 and downstream cytokine IL-1β in keratinocytes of patients with vitiligo. NLRP3 and IL-1β levels are consistently increased in perilesional skin and the IL-1β levels correlate with disease activity and severity [29], suggesting that the NLRP3 inflammasome plays a role in the progression of the disease, particularly since the NLRP3 inflammasome stimulates the T-cell response [29].

#### 2.2.5. IL-6

IL-6 is secreted by T lymphocytes and macrophages. Increased serum levels of IL-6 are reported in patients with vitiligo [52], in particular in those with new lesions [53]. Oxidative stress may influence IL-6 expression. This cytokine is overexpressed in the presence of subtoxic levels of hydrogen peroxide (H_2_O_2_) [54] and after the exposure of melanocytes to phenols, through activation of the unfolded protein response (UPR) [55]. Therefore, IL-6 might have a role in the activation of immune responses following oxidative injuries in vitiligo [55]. Moreover, a specific IL-6 polymorphism, *IL6-5*72 G/C, has been proposed to be associated with vitiligo susceptibility in the Gujarat population [56].

#### 2.2.6. IL-17

IL-17 is a pro-inflammatory cytokine with a clear connection to numerous inflammatory diseases, such as psoriasis. Serum and tissue levels of IL-17 are higher in vitiligo patients than in controls [49,53,57] and a positive correlation was observed between IL-17 levels and the duration and body surface area (BSA) of the disease [52,57]. IL-17 cytokines are produced mainly by T helper 17 (Th17) cells, which are increased in the serum of patients with active vitiligo [58] and a positive correlation was also found between circulating Th17 cells and BSA, suggesting a role in progressive disease [59]. Although these observations highlight the involvement of IL-17 in vitiligo, its role seems to be secondary compared to other cytokines. IL-17 may promote the persistence of the inflammatory network through the release of other pro-inflammatory cytokines, such as TNF-alpha, IL-1β and IL-6 by monocytic cells [60]. In autoimmune vitiligo lesions of SLC, the expression of IL-17 is low, suggesting that it might be a cytokine involved in the progression of a process triggered by other immunological factors [34].

#### 2.2.7. IL-22

IL-22 is produced by several innate and adaptive immune cells. Among the latter, Th17 and Th22 are the main source of IL-22. Serum levels of IL-22 are higher in vitiligo patients than in controls [52,61] and are significantly increased in generalized vitiligo compared to localized disease [52]. As extensively investigated by Dong et al. [62], IL-22 plays a main role in the pathogenesis of vitiligo. It may trigger keratinocytes to release IL-1β via the involvement of the NLRP3 inflammasome [62]. Keratinocytes exposed to IL-22 may be able to inhibit melanogenesis and induce the apoptosis of melanocytes [62]. The signaling of IL-22 is mediated by the JAK-STAT pathway, in particular STAT3 [62], suggesting a possible therapeutic target for future therapies.

#### 2.2.8. IL-21, IL-23, IL-15

IL-21 is upregulated in vitiligo of SLCs, and increased serum levels have been detected in vitiligo patients with a positive correlation with BSA [34,59]. IL-21 may participate in inducing IL-23 receptor expression, driving Th17 polarization [59]. IL-23 serum levels are higher in patients than in controls and positively correlate with the duration and extent of vitiligo [63]. IL-23 expression by dendritic cells may have a role in promoting inflammation in active disease, inducing the development of Th17 cells [63]. Nevertheless, both IL-17 and IL-23 serum levels showed no significant differences between vitiligo patients and controls in a Sudanese population [64]. IL-15 serum levels are higher in vitiligo patients than in controls and show a positive correlation with the extent of the disease [65]. IL-15 is an important cytokine in vitiligo pathogenesis thanks to its ability to regulate IL-17 levels and maintain signals of T cell memory (TRM) [66]. TRM cells infiltrate vitiligo skin and contribute to the maintenance of disease, producing IFNγ and TNFα and exhibiting cytotoxic activity against melanocytes [66,67]. TRM cells remain in perilesional skin after the resolution of inflammation and may stimulate a recurrence of the disease [67]. The recruitment of TRM cells into the skin may be dependent upon CXCR3-CXCL10 signaling [67].

The main mechanisms so far cited have been resumed in Figure 1.

## 3. Vitiligo Treatment Options: Past, Present and Future Therapies

### 3.1. Systemic Therapeutic Approaches and Biologics for the Treatment of Vitiligo: Rationale, Efficacy and Safety

As described so far, a great variety of factors is involved in the pathogenesis of vitiligo. Several attempts have been made by the scientific community to find an effective treatment and one of the therapeutic approaches is mainly based on the use of pro-inflammatory cytokines as targets. In this context, monoclonal antibodies have been synthesized, either derived from human immortalized cells or from other animals, targeting the key factors involved in the physiopathology of different diseases, including vitiligo [68]. For instance, several biologics have been used in order to achieve repigmentation or halt the depigmentation processes. Ustekinumab, a monoclonal antibody blocking both IL 12 and IL-23, is one of these molecules. Originally developed for psoriasis [69], this monoclonal antibody has been used in several patients. Since psoriasis is a much more common condition than vitiligo, it has been used in patients affected both by psoriasis and vitiligo. A case report demonstrated that the antagonism of IL-12/23 action led to the resolution of psoriatic plaques, along with a remarkable improvement of the concomitant alopecia and vitiligo in a patient affected by psoriasis, alopecia areata and vitiligo at the same time. Therefore, it has been hypothesized that cases of vitiligo that are resistant to common treatments could benefit from the use of this antibody [70]. Unfortunately, this case report is the only positive result regarding the use of ustekinumab in vitiligo. One nation-wide study, in fact, demonstrated the new onset of white patches in three patients treated with ustekinumab, thus worsening the pre-existing vitiligo in one more patient [71]. In another study, 15 patients developed vitiligous patches after ustekinumab treatment [72] and another case report described a similar reaction [73]. However, the relationship between ustekinum abuse and the appearance of vitiligo patches should be further investigated. Secukinumab is another tested monoclonal antibody (MAB) that may be considered as a suitable candidate for vitiligo treatment. This biological drug targets IL-17A, an interleukin involved in the pathogenesis of vitiligo, psoriasis and psoriatic arthritis. Again, the studies published to date have led to contrasting results. One case report from 2020 revealed that adalimumab, a TNF-alpha inhibitor, failed in treating psoriasis, but this patient also showed an onset of vitiligo lesions over his body. After the switch to secukinumab, an improvement of the psoriatic lesions and the clearance of vitiligo patches were observed [74]. On the other hand, two studies denied this possibility. In a 2020 case report, two patients who failed a previous systemic treatment for psoriasis switched to secukinumab and vitiligo-like patches appeared over their skin; vitiligo was treated as a side effect with topical tacrolimus, stopping the progression of previous patches [75]. The other study was a 2019 clinical trial. Eight patients were enrolled to challenge the efficacy of IL-17 inhibition but unfortunately most patients developed more lesions than when they started (7/8 patients) and the study was interrupted. Although the trial was prematurely interrupted, the hypothesis of the shift towards the Th1 profile of the lymphocytic population and the intermediate step of Th17.1 was imagined to be the crucial pathogenetic event [76]. New-onset vitiligo may occur as paradoxical skin reaction during treatment with TNFα-IL-17 and IL-12/IL23 inhibitors in patients with other inflammatory diseases [71]. A retrospective study showed that new cases of vitiligo were related to infliximab and adalimumab use (72.2%) and to ustekinumab and secukinumab (22.2%) [71]. However, adalimumab was the major trigger of new-onset vitiligo [71]. The maintenance of the biological therapy after the onset of vitiligo led to stable disease or repigmentation in the majority of cases, whereas the use of biological agents in patients with pre-existing vitiligo had an unfavorable outcome, with vitiligo progression in 43.7% of cases and repigmentation in only one out of eighteen patients [71]. The onset of vitiligo during the treatment with adalimumab [74,77,78,79,80,81] and infliximab [82,83,84,85,86,87,88] for other inflammatory diseases has been widely reported in the literature. The occurrence of vitiligo in patients treated with anti-TNFα has limited their use, although they appear hopeful in stopping the progression of the disease. In a group of six patients treated with anti-TNFα, five patients did not develop any new depigmented patches, although repigmentation was not observed [89]. Some authors suggested that anti-TNFα may be useful to stabilize vitiligo rather than to favor repigmentation [90]. The results are inconclusive about the use of etanercept, another TNFα inhibitor, in patients with vitiligo, and the low number of patients treated with this biological agent does not allow us to give a final opinion. Etanercept led to a mild improvement of vitiligo in two patients who showed strong cytoplasmic staining for TNF-α in the samples obtained from their margin lesions [91] and in a patient affected by both vitiligo and psoriasis [92]. Nevertheless, an open-label pilot study treatment with etanercept in four patients with vitiligo reported neither improvement nor aggravation of the disease [93]. Hence, the authors did not recommend etanercept as a first-line treatment for vitiligo [92]. A more promising therapy seems to be tofacitinib, a Janus kinase (JAK) inhibitor already used in several inflammatory-mediated diseases. Several studies have been proposed, mostly pilot studies, both paired with and without narrow-band UVB therapy (NB-UVB). Scarce results were observed in the improvement of vitiligo lesions with the use of oral tofacitinib alone [94], two case reports also demonstrated a partial and progressive resolution of depigmentation in a patient treated with oral tofacitinib [95,96] and another pilot study with 16 patients using topical tofacitinib showed positive results, especially in darker skin types and younger patients [97] and one more patient treated with increasing doses of oral tofacitinib after failing UVB therapy [98]. Although all these studies have reported positive results regarding this therapy alone (not paired with NB-UVB), not just regarding the progression but also the re-pigmentation of previously affected areas, confounding factors persist, such as previous NB-UVB therapy and the scarcity of numbers, preventing researchers from drawing any conclusion. On the other hand, several studies have evaluated the combination of phototherapy and tofacitinib together. One of these studies suggests that NB-UVB therapy is mandatory for re-pigmentation to let the melanocytes repopulate affected areas, whereas tofacitinib acts as the immune system inhibitor [99]. This consideration was further proposed by another case report that described a male patient which improved both his vitiligo, alopecia areata and psoriasis lesions after oral tofacitinib and UVB therapy in a three-month period [100]. Finally, a retrospective study with a more consistent number of patients affected by vitiligo reported an almost complete re-pigmentation of vitiligo patches (92% patients) following oral tofacitinib and UVB therapy, compared to patients treated solely with UVB therapy (77% re-pigmentation rate) [101]. All these studies can be summarized by the final considerations of the model proposed by Liu et al.: tofacitinib is a safe drug to use, with only mild side effects, such as plasmatic levels of lipids growing or some moderate weight gain. Both the immunosuppressive and repopulating effects derived from tofacitinib require light exposure, especially for re-pigmentation; oral tofacitinib alone might be useful in monotherapy just for maintenance [102]. On the downside of this treatment, the high cost of this therapy and the chance of developing leukopenia and secondary infections might be considered as two important limitations to the use of this therapeutic approach on a large scale. In fact, a topical administration has been mainly considered, for visible areas such as in the face [103], and in children as well [104], with positive results. Recently, ruxolitinib, a JAK1/2 inhibitor, has been proposed for the treatment of vitiligo. Although JAK1/2 inhibition seems to be a better treatment option compared to JAK1/3 inhibition by tofacitinib, the results obtained from a randomized trial [105] showed comparable effects, whereas other case reports [106,107] have confirmed a lack of significant improvement. As mentioned, IL-6 is implicated in the immunopathogenesis of vitiligo, in particular in active disease [53]. Tocilizumab is an anti -interleukin -6 (IL -6) receptor humanized monoclonal antibody, approved for the treatment of rheumatoid arthritis and juvenile idiopathic arthritis (JIA). Nadesalingam et al. reported the onset of vitiligo, halo naevi and alopecia areata during treatment of JIA with tocilizumab [108]. This effect is the consequence of the blockade of the IL-6 receptor that inhibits the interaction of IL-6 with its receptors, thus increasing IL-6 levels in serum [108]. Moreover, the increase in IL-6 levels during treatment with tocilizumab may lead to melanocyte damage [108]. In contrast, tocilizumab was used in a patient with seronegative rheumatoid arthritis and the almost complete resolution of simultaneous lesions of vitiligo was observed; nevertheless, the recurrence of depigmentation at the original sites has been noted after the discontinuation of tocilizumab and the treatment has not resolved genital vitiligo [109]. All these possible targets have been summarized recently in a review about biologic-induced vitiligo, showing that we still lack sufficient knowledge to address the reasons why a drug targeting a specific cytokine involved in the pathogenesis of this condition leads to paradoxical results. Table 1 presents a short summary of the most common attempted treatments.

### 3.2. Perspectives to Explore

So far, biological therapies have been attempted for several skin diseases but in many cases, the first steps of the research have been characterized by many failed attempts and the appearance of vitiligo as a side effect [110].

Although the biological agents discussed so far have brought about some promising results, the scientific community continues to evaluate new therapeutic frontiers, exploiting new therapeutic targets. In this context, other molecular pathways have been studied using in vitro and in vivo experimental approaches. Anti-IFNγ antibodies were administered with intradermal injections around the lesion or through intramuscular injections, but the number of treated patients was very small (four in total), two of which responded with repigmentation if the drug was administered locally and the other two required, for a response, a subsequent intramuscular administration [111]. In another study, the researchers reported that IFNγ induces the accumulation of melanocyte-specific CTLs in the lesional skin and vitiligo mice treated with anti-IFNγ antibodies for 5 weeks, stopped the depigmentation and reduced the number of melanocyte-specific CTLs [112]. IFNγ activity is mediated by the JAK-STAT pathway, the inhibition of which by biological agents leads to re-pigmentation via the immunosuppression and repopulation of the melanocytic cells. The CXCL10-CXCR3 axis is an IFN-induced pathway, and its neutralization has been investigated in a mouse model [41]. Mice with established depigmentation treated with CXCL10 neutralizing antibodies for eight weeks showed reversal of the disease with repigmentation [41]. The use of these treatments, such as tofacitinib and ruxolitinib, as previously mentioned, produced the most consistently positive results in terms of efficacy and safety. The inducible form of the NO synthase (iNOS) may be considered as a possible target for future therapies since it is regulated by IFN, IL-33 and IL-6 and might play a role in the stress-related cell damage in vitiligo skin. However, to date, no valid study has been focused in this direction, but this target could be exploited after acute phases of the disease, as a maintenance treatment. T_RM_ cells may infiltrate perilesional skin and are able to trigger vitiligo after the cessation of treatment [113]. Indeed, IL-15 is required for the maintenance of T_RM_ cells in the skin [66]. CD122 is a component of a subtype of the IL-15 receptor, which is expressed on memory T cells [113]. An anti-CD122 antibody has been used to target IL-15 receptors; its blockade has reversed the disease in mice with stable vitiligo [113]. A similar experiment has been performed to stop the recruitment of recirculating memory T cells (T_CM_) in a vitiligo mouse model [114]. Naïve lymphocytes need to express sphingosine-1 phosphate receptor 1 (S1PR1) and to detect its ligand S1P to access the circulatory system. When entering tissues, memory T cells downregulate S1PR1 to prevent recirculation [115]. In a study by Richmond et al. (2019), the inhibition of S1P in mice reversed vitiligo, indicating that T_CM_ cells cooperate with T_RM_ to maintain disease [114]. Other studies regarding T-reg cells have proposed the use of programmed cell death 1 (PD-1) fusion protein as a possible therapeutic option, considering that T-reg cells play a pivotal role in the pathogenesis of vitiligo. The authors of this study demonstrated a >50% area of repigmentation in this murine model up to 8 weeks of treatment after the last injection, comparing it with simvastatin oral administration, which led to a <1 repigmentation response [116]. As stated in another study, in fact, simvastatin may have a rationale in the treatment of vitiligo since hydroxymethylglutaril-coenzyme A (HMG-CoA) inhibition leads to the stopping of IFN production and CD8+ proliferation, but greater numbers are needed in order to propose this treatment as an option for future therapies [117]. Altogether, these studies suggest the need for further clinical trials to achieve enough enrolled patients which can undergo this type of treatment. Table 2 produces a short resume of future therapeutic perspectives.

## 4. Conclusions

The pathogenesis of vitiligo involves several pathways. Previously submitted treatments attempted to suppress the immune system without aiming at a proper target. In time, with the advent of biologics, the latest therapies have focused on specific targets and objectives that might produce satisfying results in terms of safety and effectively stop disease progression. To our knowledge, it can be advisable to compare biologics with other new treatments for the management of the complex cytokine network and combined therapy might be a future approach, especially physical treatments such as NB-UVB therapy and excimer lasers. Combined therapy could prove useful in terms of achieving longer disease-free periods of time or rapid repigmentation.

## Figures and Tables

**Figure 1 ijms-22-11429-f001:**
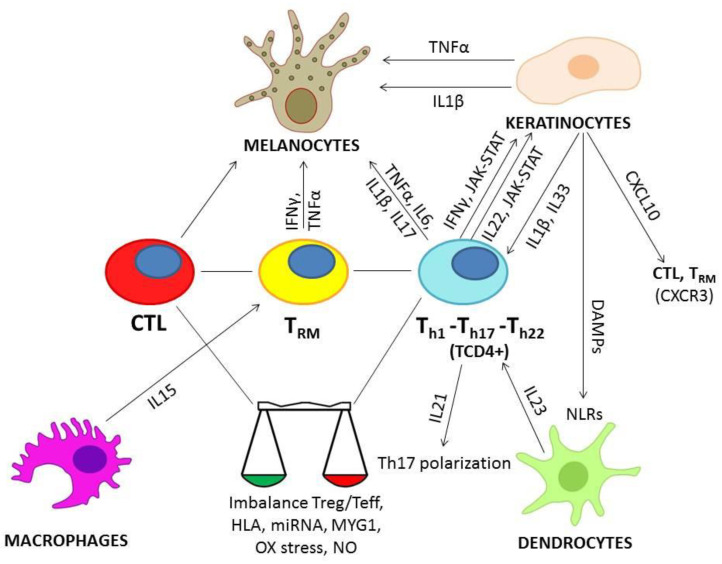
Role of the main cellular populations and interleukins involved in the pathogenesis of vitiligo. Arrows are matched with related pathways/interleukins primarily involved in the interactions between cells.

**Table 1 ijms-22-11429-t001:** Biological drugs used for the treatment of vitiligo.

	Target	Desired Effect	Real-Life Effect
Ustekinumab [69,70,71,72,73]	IL-12 and -23	Blocking of inflammation and Th17 polarization	Appearance of new vitiligo patches
Secukinumab [74,75,76]	IL-17A	Interrupting inflammation and production of other proinflammatory cytokines	Progression and appearance of new depigmentation areas
Adalimumab [77,78,79,80,81,89,90], infliximab [82,83,84,85,86,87,88,89,90], etanercept [91,92,93]	TNF-alpha	Stopping the progression of inflammation	Contrasting results in the appearance of new patches and the progression of already existing ones
Tildrakizumab	IL-23	Blockage of the inflammatory network	Insufficient studies
Tocilizumab [108]	IL-6 receptor	Stopping the propagation of inflammation	Soluble form of IL-6 might be causative of new manifestations
Tofacitinib [94,95,96,97,98,99,100,101,102,103,104], ruxolitinib [105,106,107]	JAK1-3 and 1-2	Halting inflammation cascade signals	Stopped progression. Might need concomitant UVB for repigmentation

**Table 2 ijms-22-11429-t002:** Possible targets for the treatment of vitiligo, known effects on melanocytes, molecules related to the objectives, scientific rationale.

	Involvement	Possible Targets	Rationale
Pd-1	Immunity response and checkpoint function	PD-1, PD-L1	Regulating T-cell activation
IFN-gamma	Inflammation and promotion of autophagy	IFN-gamma soluble, CXCL10-CXCR3	Stopping specific CTLs killing of melanocytes
NOS	Production of oxygen radical species	Inducible synthase (iNOS)	Lower levels of oxidative stress
IL-15	Regulates level of IL-17	Soluble form and receptor CD122	Stopping crosstalk between T_RM_ cells and Tcm cells
S1PR1	Transit from tissues to blood vessels of T lymphocytes	S1PR1 (receptor) or S1P (ligand)	Allowing the recirculation of T memory cells and preventing the maintenance of inflammation

## Data Availability

Not applicable.

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
