# Peer review of "Role of Cytokines in Vitiligo: Pathogenesis and Possible Targets for Old and New Treatments"

_ijms, 2021, doi:10.3390/ijms222111429_

Round 1

Reviewer 1 Report

The paper presented for review deals with the immune pathomechanisms governing the development of vitiligo. It describes the contribution of various cytokines and presents recent updates on treatment options. The paper is well-structured, comprehensive and elegantly written. Although the sections “Results” and “Discussion” should be renamed for the manuscript to more resemble a review rather than an original paper. In my opinion the paper does not present any major flaws or concerns. Although I list below some of my suggestions and very minor corrections.

In the IFN-γ section the authors state that it inhibits melanogenesis, but vitiligo is characterized by melanocyte loss from the epidermis rather than mere pigment loss. It would be worth mentioning that an alternative mechanism exists that explains the role of this factor in vitiligo and that is induction of apoptosis by CD8+ T cells. 

The authors might want to discuss the PD-L1 (CD274) glycoprotein as promising target for vitiligo treatment: Miao, X., Xu, R., Fan, B. et al. PD-L1 reverses depigmentation in Pmel-1 vitiligo mice by increasing the abundance of Tregs in the skin. Sci Rep 8, 1605 (2018). https://doi.org/10.1038/s41598-018-19407-w

The authors may want to mention a clinical trial on a STAT1 inhibitor simvastatin: Zhang S, Zdravković TP, Wang T, et al Efficacy and safety of oral simvastatin in the treatment of patients with vitiligo Journal of Investigative Medicine 2021;69:393-396.

The authors might want to consider discussing other non-pharmacological therapies such as excimer lamps and laser and topical latanoprost cotreatment

Please decipher abbreviations like NLRs, Mab

Author Response

Dear reviewer, thank you for your kind responses and observations. Here a point to point list of answers:

1) results and discussion have been renamed accordingly;

2) in the IFN section, a few lines have been added for the specific apoptosis link and CD8+

3) & 4) articles were added as suggested in the formerly known chapter "Discussion" and the relative treatment added in Table 2

5) a few lines were added in the conclusion section regarding excimer lasers and uvb therapy but it was the authors' opinion that adding too much detail regarding latanoprost or other physical treatments would derail from the original topic at hand

6) proper description of abbreviations were added.

Thank you again.

Reviewer 2 Report

Paolo Custurone et al presented a review about the role of cytokines in vitiligo.

It is a meaningful work to review and summarize the cytokines which have been reported involved in vitiligo.

Here, only minor suggestion.

  1. Could you please rewrite Figure 1, many information were not mentioned in this figure, for example, IL21, DAMPs-NLRP in keratinocytes, CD4 T cells….
  2. It should be much better if the no. of references about the clinical reports was added in Table 1.

Author Response

Dear reviewer, thank you for your interesting observations. Here a point to point list of answers:

1) the suggested lymphcytes and cytokines have been added accordingly to Figure 1, in order to better define our research findings;

2) citations have been added in the Table 1 in order to make the paper more fruible.

Thank you again.